# A Novel Approach to Discrete Representative Volume Element Automation and Generation-DRAGen

**DOI:** 10.3390/ma13081887

**Published:** 2020-04-17

**Authors:** Manuel Henrich, Felix Pütz, Sebastian Münstermann

**Affiliations:** Integrity of Materials and Structures (IMS), Department of Ferrous Metallurgy (IEHK), RWTH Aachen University, Intzestraße 1, 52072 Aachen, Germany

**Keywords:** representative volume elements, microstructure, random sequential addition algorithm, voronoi tessellation, discrete tessellation

## Abstract

In this study, a novel approach for generating Representative Volume Elements (RVEs) is introduced. In contrast to common generators, the new RVE generator is based on discrete methods to reconstruct synthetic microstructures, using simple methods and a modular structure. The plain and uncomplicated structure of the generator makes the extension with new features quite simple. It is discussed why certain features are essential for microstructural simulations. The discrete methods are implemented into a python tool. A Random Sequential Addition (RSA)-Algorithm for discrete volumes is developed and the tessellation is realized with a discrete tessellation function. The results show that the generator can successfully reconstruct realistic microstructures with elongated grains and martensite bands from given input data sets.

## 1. Introduction

Most modern steels utilize different phases to improve their mechanical properties for their use case. A family of steels that makes use of that principle are the Advanced High Strength Steels (AHSS). A popular example for a steel grade from this family are dual-phase steels, which, concerning the strength, have been steadily improved in the past decade [1,2]. Yet, while the combination of hard and soft phases has a very valuable influence on the hardening behavior and the ultimate tensile strength of the material, the phase boundaries also lead to damage initiation [3]. Especially when it comes to fatigue properties the presence of multiple phases becomes a complicated circumstance. In regular single phase steels the main reason for the initiation of fatigue cracks are said to be non metallic inclusions [4]. However, a hard martensite phase that is enclosed by a soft ferrite matrix can adapt this behavior of non metallic inclusions and therefore initiate fatigue cracks as well [5]. Studying the influence of inclusions on fatigue life has already been a complex task and several articles and books have been published on this specific topic [6,7,8]. The influence on fatigue of a second phase however, was described much less not to speak of the combination of non metallic inclusions and a second phase [9]. The reason for this is the fact that an experimental study of the influence of interactions between multiple phases is very complex and requires sophisticated and expensive measurement techniques. Not only is the measurement of the interaction complicated to realize but also some of the parameters within a microstructure are very complicated to control. Yet, a systematic variation within a microstructure can be necessary for a detailed description of the microstructure behavior under loading [10,11].

Since the experimental characterization of these properties is quite a complex task, methods in the field of computational simulation have been developed as a different approach. Within this field a strategy was introduced to describe the materials behavior. It is called multilevel finite element method (FEM) or FE${}^{2}$ method. This method works on two levels, the macro-level which applies macroscopic models of continuum mechanics and the micro-level which is based on constitutive relations often combined with microstructure models. These models directly contain geometrical and crystallographic information about the real microstructure. Material models that can be used with the FE${}^{2}$ method are for example, the Modified Bai–Wierzbicki plasticity model (MBW) as described by Lian et al. and Wu et al. [12,13]. Another type of Material models are Crystal Plasticity models (CP) as they were used by for example, Cruzado et al. or Segurado et al. [14,15]. A quite detailed summary of other constitutive laws, kinematics, homogenization and multiscale methods is given by Rothers et al. [16].

Two approaches of modeling the microstructure exist: Modeling the real microstructure based on microstructure analysis and generating synthetic microstructures based on statistical data. Real microstructure models are more or less a digital copy of microstructure data that was captured via any method that allows a visualization of the microstructure. A more sophisticated method is the use of Representative Volume Elements (RVEs). Many frameworks used for FE${}^{2}$ method use Representative Volume Elements for the micro scale level [17,18].

RVEs in this case are small volume elements that represent the microstructure on a statistically profound basis. However, there are several requirements for the generation of RVEs. A summary of these requirments can be found in the work of Gitman et al. or also in the paper of Pelissou et al. [19,20]. Gitman draws the conclusion that all these requirements have in their core the same message, that RVEs “should contain enough information on the microstructure yet be sufficiently smaller than the macroscopic structural dimensions. Thus, a separation of scales should be enabled” [19].

Therefore, it is necessary to extract a database for the RVE generation from microstructure analysis. There are several methods to extract the database from the microstructure. For 2D RVEs it is very simple, since pictures from light optical microscopy (LOM) or Scanning Electron Microscopy (SEM) deliver the desired data already. The 3D case however is much more complex and more sophisticated methods need to be used in order to characterize the microstructure. Bargmann et al. describe two major methods how to generate 3D data for a microstructure analysis. The serial sectioning technique where the microstructure is scanned slice by slice and after scanning the slices can be put together and the projection based imaging where the sample rotates inside a radiation beam and the microstructure can be reconstructed after a series of scans [21]. Another method how to generate a 3D data set is to take pictures in three different directions. This method is not quite as accurate as the methods mentioned above. However, it is very fast and much cheaper. Therefore this is often the method of choice for a first guess of the 3D microstructure. The pictures can be taken from a LOM or from an SEM. The advantage of the SEM is that there are numerous detectors that can be used with it. For example, the Electron Backscatter Diffraction (EBSD) whose data can be assessed by the MATLAB (Matlab R2019a—The Mathworks Inc., Natick, MA, USA) toolbox MTEX (MTEX version 5.3.0) [22]. In order to get a data set that is statistically representative these measurements have to be done repetitively. The microstructure’s parameters of all measurements can then be used for statistical analysis [23]. For instance, the grain size of one phase in the steel can be described by a distribution function. Other parameters that are interesting for generating RVEs of steel microstructures are the aspect ratio of smallest and biggest radius in each grain, the grain orientation, the slope describing the angle between the main axis of the grain and the axis of rolling direction and banding of secondary phases such as martensite bands that appear in some DP steel grades. The MTEX toolbox can analyze EBSD data in terms of grain size, shape, slope and orientations. When MTEX is utilized a csv-file is generated containing the data of interest. Such kind of csv-file was used as input data for the RVEs investigated in this study. The focus however, will lie on the generation of the RVE itself using the input from MTEX as exemplary data [22].

For the generation of RVEs representing polycrystalline microstructures, several steps must be followed:Generation of spheres or ellipsoids in dependence of a distribution function from the statistical characterizationPacking of the grain representing volumes (spheres or ellipsoids)Tessellation to fill empty spaces between the volumesAssignment of crystallographic orientation

In most popular RVE generators like Dream.3D (DREAM.3D Version 6.5, BlueQuartz Software, Springboro, OH, USA) or Neper (Neper version 3.5.2) these steps are realized in some way. Dream.3D for example runs the statistical analysis of the EBSD measurement in its own framework. Several filters are given that can be used in order to investigate and evaluate EBSD data. The user can build so called pipelines by connecting filters and thus generate the desired input data for an RVE. The RVE generation itself follows exactly the above mentioned steps. The packing of the volumes is in this case solved with a quite complex grain packer using neighborhood distributions. It is called the Constrained Grain Packer (CGP) because the packing is completely controlled. The following tessellation in Dream.3D is solved with a voronoi tessellation. However, one voronoi cell does not necessarily represent one grain. It is certainly possible that several voronoi cells represent one single grain. This strategy solves the problem how to generate concave shapes with an only convex growing algorithm. Yet, again this approach is already in its idea quite complex. At the end each grain will be assigned a crystallographic orientation [23,24].

Neper on the other hand does not make use of the packing. Neper generates its RVES with a multiplicatively weighted voronoi tessellation with randomly distributed seeds based on grain size distributions. Also, multiscale tessellation is possible with Neper. This feature can be useful when primary phases need to be divided in subdivisions in order to change properties. However, the approach does not consider elongated or concave shaped grains. Therefore complex microstructures with elongated grains are difficult to reconstruct with the methods used by Neper [25].

Most generators, like the two mentioned above make use of the Voronoi tessellation for a good reason. It delivers an analytical description of a polyhedron which can be used to approximate grains in polycrystals. The mathematical description of a Voronoi-Polyhedron is defined as follows:(1)$$\begin{array}{c} C_{i}=\left\{P\left(\vec{x}\right)\epsilon D|d\left(P,G_{i}\right)<d\left(P,G_{j}\right)\forall j\neq i\right\}, \end{array}$$
where $D\in {ℜ}^{n}$ is a spatial domain, *D* contains a set of Points $\left\{G_{i}\left(\vec{x_{i}}\right)\right\}$ and d(•,•) is a norm [24]. Yet, this polyhedron is still an approximation of a sphere. As soon as the microstructure of a polycrystal does not show spherical grains the voronoi tessellation however, is not the right tool anymore. Dream.3D does have a solution to describe non spherical shapes with the voronoi tessellation yet, only with huge effort and a very complex method.

The RVE-Generator described by Vajragupta et al. also follows the above mentioned steps. The packing method however was realized with a Random Sequential Addition (RSA) algorithm considering spheres. Under these circumstances the voro++ package from Rycroft [26] which contains a multiplicatively weighted Voronoi algorithm could be used for filling the empty spaces between the spheres. After generating the grain geometry python scripts are used for meshing and generating an FEM-Model to be calculated in Abaqus.

The authors of this paper have continuously been working on RVE generators in order to find the best methods to reconstruct the microstructures of interest. The main focus here lies on a the appropriate description of the real microstructure without generating an exact copy. Recent studies have shown that the influence of local effects such as inclusions, grain morphology, banding of the microstructure do have a significant influence on local events like the initiation fatigue cracks, ductile damage or brittle fracture initiation [4,9,27,28]. Also features like periodicity play an important role in order to keep the RVEs representative [19,29]. All of the desired features can be realized with given generators. Yet, none of them offers a combination of all features at the same time. Therefore, it is desired to have an RVE generator being capable of combining all the features of interest and reconstructing microstructures based on statistical analysis as similar to the real microstructure as possible. Since real microstructure models cannot be used for optimization issues or microstructure design they are not favored. Thus, the following main problems were identified that need to be addressed in a new generator:Spheres are not suitable as initial geometry for grains in a metallic polycrystal.Ellipsoids or even more complex bodies must be used as initial geometry.Tessellation must support a growth that keeps the initial geometry.Banding of second phases has to be considered during the modeling of the microstructure.Inclusions should be considered as well.RVEs must be periodic in order to represent the continuous space.

Since the number of required features is quite high the goal of this study is to reach the best possible results with the simplest possible methods. The key feature to reach this goal is a module containing a discrete tessellation function which can be controlled easily and also works with in an intuitive way, which allows additions of other features to be easily implemented. The strategy is based on a modular approach in order to be able to switch features on and off at all times.

## 2. Materials and Methods

In this section a short description of the material is given. Yet, the focus will lie on the methods how to convert a given set of input data into a three dimensional microstructure model with the required features mentioned in Section 1. Thus, aspects of data acquisition will not be discussed. The approach of the discrete RSA and Tessellation will be explained.

### 2.1. DP800

The material used for this study is a DP800. This steel has a dual phase matrix containing ferrite and martensite. The microstructure is shown in Figure 1.

Figure 1a can be used to determine the phase fraction of ferrite and martensite with a gray scale analysis. For this material the fraction is found to be 30/70. Information about the texture, grain size, grainshape and slope can be determined with EBSD pictures as shown in Figure 1b. The input data that is used for the generation of the RVEs in this study was generated with the matlab toolbox MTEX [22]. Although the MTEX analysis was carried out with the greatest care, not too much emphasis is placed on the accuracy of this data, as it is only used to check the methods used. Only when the generator is validated and the output matches the input within an acceptable range is it relevant whether the input data describes the material of interest well enough. The features considered in this study are the grain size and shape.

Figure 1a shows a microstructure of a DP800 steel. The nonindexed area, depicted by white color is assumed to be 100% martensite. The picture clearly shows that the secondary phase sometimes shows banding within the microstructure. These bands have a significant influence on the mechanical properties of the material and behave differently from martensite islands. Therefore they also need to be considered when modeling a synthetic microstructure [3,28].

Figure 2 shows the log probability density which was evaluated with a Kernel Density Estimation (KDE) for the grain size (Figure 2a) and the aspect ratio (Figure 2b). It is important to mention here that the KDE method is to be much more trusted than the common method using histograms and regular log-normal or gamma distributions. These methods are highly dependent on the bin size of the histogram and thus also the log normal distribution, which is fitted to the histogram. Therefore, histograms should always be considered with great caution. KDEs on the other hand deliver very robust values since every data point is weighted with a Gaussian distribution on its own [30]. The KDEs were calculated with the data that was extracted from the EBSD with the MTEX toolbox. The aspect ratio distribution shows that spheres, having an aspect ratio of 1 do not represent the grain shape of this microstructure. Therefore, ellipsoids seem to be the right choice. The probability density of the grain size could not directly be used as input for the RVE-Generation because it is measured via the grain area that is visible on EBSD-pictures. For a valid reconstruction of a 3D-volume 3D-input data is necessary. Therefore the radii of the EBSD-measurement were taken and ellipsoids were interpolated in order to generate input data for the RVE-Generation. For the first approach only a single phase will be considered for generating the RVE. Once it is shown that the generator works for one phase it is quite simple to add a second phase with its own distribution. Therefore, the martensite phase will be neglected in the following sections. However, Section 2.2.2 explains the consideration of the martensitic phase and the construction of martensite bands.

### 2.2. RVE-Generation

In order to generate RVEs in agreement with the definition mentioned in Section 1 a strategy was chosen in which several steps build on each other in a modular way. The steps are listed below:Generation of input dataPlacing ellipsoids in volume (RSA)Filling empty spaces between ellipsoids (Discrete Tessellation)Validation of output with input

In this study an exemplary input has been chosen to demonstrate how the generator works. The ellipsoids representing early versions of the grains were placed into the volume with a discrete version of the RSA algorithm. The RSA algorithm leads to empty spaces between the ellipsoids which have to be filled with material. This is done with a discrete tessellation module that lets every ellipsoid grow stepwise into the free space. After the RVE has grown to its final stage the generated data also called output data must be validated with the input data, which is done with another module. The generator is written in pure python code. Therefore it is in its nature already build up in a modular way and other modules that add more features to the RVE can easily be implemented. It must be noted here that the goal of the generator is to reconstruct a target microstructure. In the following point clouds will also be referred to as grains and the growth of point clouds will be called grain growth. These expressions must not be confused in any sense with metallurgical grain growth or similar. The following description refers only to point clouds developing into their target shape and target size in order to depict the grains of the target microstructure correctly.

#### 2.2.1. Discrete RSA Algorithm

The input data for the RSA algorithm contains the information about every grain that is supposed to be reconstructed. The grains are approximated by ellipsoids and the sum of all ellipsoid volumes can be considered as the RVE volume. However since it is not possible to reach a packing ratio of 100% with ellipsoids even if they differ in size they have to be manipulated before being placed into the volume. Therefore, every ellipsoid is shrinked down to for example, 20% of its initial volume. Thus, the sum of all ellipsoid volumes is 20% of the RVE volume and the packing ratio after the RSA algorithm has finished is 20%. This is very easy to accomplish and therefore speeds up the process of placing the ellipsoids into the volume. The empty space between all ellipsoids will then be assigned by the discrete growth function described in Section 2.2.2.

Unlike most RSA algorithms this version does not actually use analytic equations to detect intersections between the volumes being placed into the cube. While other versions use a continuous volume, in this case a regular cubic grid represents the volume. Therefore, three lists are initialized in the beginning of the algorithm, the Grid list, containing all Grid points at all times to generate the ellipsoids, the Trial list containing all possible midpoints for the current ellipsoid and the Taken-Points list containing all points that have been assigned to an ellipsoid already. In the beginning every point in this grid is a potential midpoint of a single ellipsoid. It is important to understand here that the points really are just points. They do not contain any volume at all, which causes a difference between the continuous volume of an ellipsoid and the discrete volume of an ellipsoid which is represented by a number of points. Each time an ellipsoid is placed in the grid all points that find themselves inside of that ellipsoid are then deleted in the Trial list and added to the Taken-Points list. Yet, before the placement of the ellipsoid is accepted all grid points within the temporary ellipsoid are compared to all the points in the Taken-Points list. If less than 1% of the points within the temporary grid are equal to any other point in Taken-Points the ellipsoid is accepted and placed into the volume. If that condition is not satisfied the ellipsoid is discarded. The size of the allowed intersection of 1% was introduced into the code to reach a higher packing ratio. It is the smallest size with an acceptable result of time savings within the RSA algorithm. The choice of midpoints is random. The size of the ellipsoids should decrease with every ellipsoid in order to avoid intersections. To make sure that is the case the .csv-file used as input is sorted by the volume of the ellipsoids. Whenever an ellipsoid is discarded only the midpoint is deleted from the Trial list. That Trial list is reinitialized whenever an ellipsoid was accepted in order to bring back the discarded midpoints which have to be considered as possible midpoints for smaller ellipsoids. The algorithm is also shortly described in Algorithm 1.
**Algorithm 1:** Discrete Random Sequential Addition Algorithm
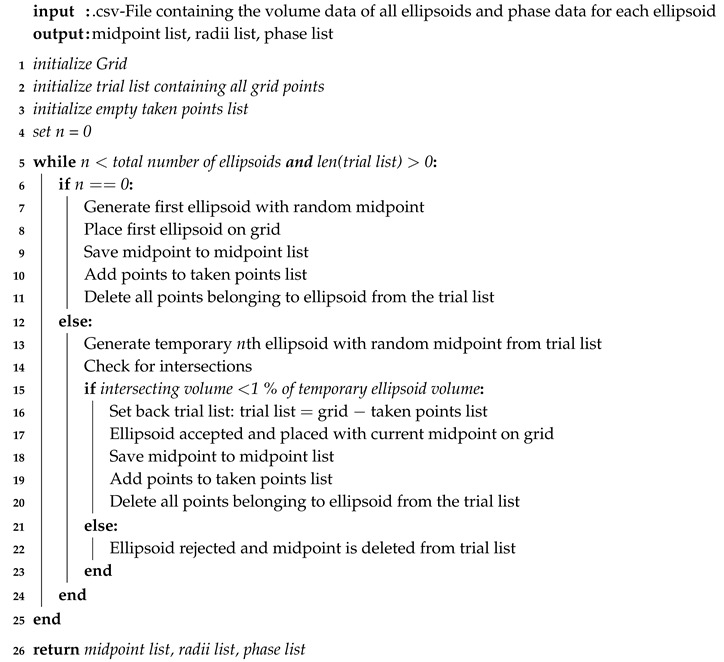


In Figure 3 two stages of the RSA algorithm are displayed. Figure 3a shows two intersecting point clouds. The red marked points are the ones intersecting. In Figure 3b the volume is displayed after all ellipsoids have been placed in the volume.

#### 2.2.2. Discrete Tessellation

The RSA algorithm returns three lists which are passed on to the tessellation algorithm. For the tessellation a grid is initialized again just like in the RSA algorithm. Afterwards all grains are rebuild from the midpoint list and the radii list. At this point the algorithm has basically reconstructed the final product of the RSA algorithm. There are several reasons for reconstructing the geometries. One could think it would be elegant to add the tessellation right after the RSA which would make the reconstruction unnecessary. Yet, in the opinion of the authors the opposite is the case. Storing the data on the way between two steps makes the whole generator more flexible in terms of further development and adding new features. Additionally, the code becomes more readable when certain modules are kept as short as possible. Another important point for the authors was the ability of debugging. The separation of the tools also creates a checkpoint where debugging can take place very easily.

Once the reconstruction of the grains is finished the algorithm can start with the actual tessellation. Since the whole volume is represented by a grid the tessellation is performed on a basis of voxels. In other words the three different radii of each grain are increased gradually. While increasing the radii, an intersection detection is performed for every grain. Once an intersection is detected the grain stops growing at that position, that no voxel can be occupied by two grains at the same time. This process runs until every voxel is assigned to one of the grains and the packing ratio of the grid is 100%. Also, every bit small grains grow is a big difference percentage wise compared to big grains.

To ensure, that the small grains do not become too large a volume control mechanism is applied parallel to the growth. So, whenever a grain reaches its predefined maximum volume the growth function neglects this grain for all further steps. In case there is some space left in the volume because some grains stopped growing before they reached their maximum volume, there is another control mechanism. This control mechanism fills up the last empty spaces randomly with the surrounding grains. Algorithm 2 shows the general structure of the tessellation. In line 19 of Algorithm 2 the RSA grain list and the index list are both randomly shuffled. It is made sure that both are shuffled in exactly the same way, to make sure the indexes still belong to the correct grain. The random shuffling is necessary to ensure that on average all grains grow simultaneously. If they grow in a certain order it appears that some grains grow around other grains because the first grain always comes first to occupy the next voxel. If there is no certain order for growing there is no order which grain comes first.

Some microstructures show so called banding in the secondary phase, for example, martensite bands that exist in DP steels due to the manufacturing process. These bands, however, should not be treated as single grains or martensite islands since the focus lies on a accurate reconstruction of the microstructure. In order to investigate local events like fatigue crack initiation caused by local properties of the microstructure, martensite bands also must be modeled separately to be able to define needed failure criteria only in the region of the martensite bands.

Here, an additional module was added to the code described above to define a complete block within the grid to represent the band. Afterwards during the RSA the band is allowed to give away its points to grains, as long as it still contains a defined percentage of the initial number of points. In other words, if a band must contain at least 60% of martensite a predefined band block with initially 1000 points can lose up to 400 points to grains during the RSA. This loss is implemented to give the band a characteristic shape that fits to the surrounding grains instead just considering it as a hexaedral block. After the RSA has finished the band is a fixed volume. It does not grow in the tessellation module. In this module the band is treated as a taken volume from the very beginning.
**Algorithm 2:** Discrete Random Sequential Addition Algorithm
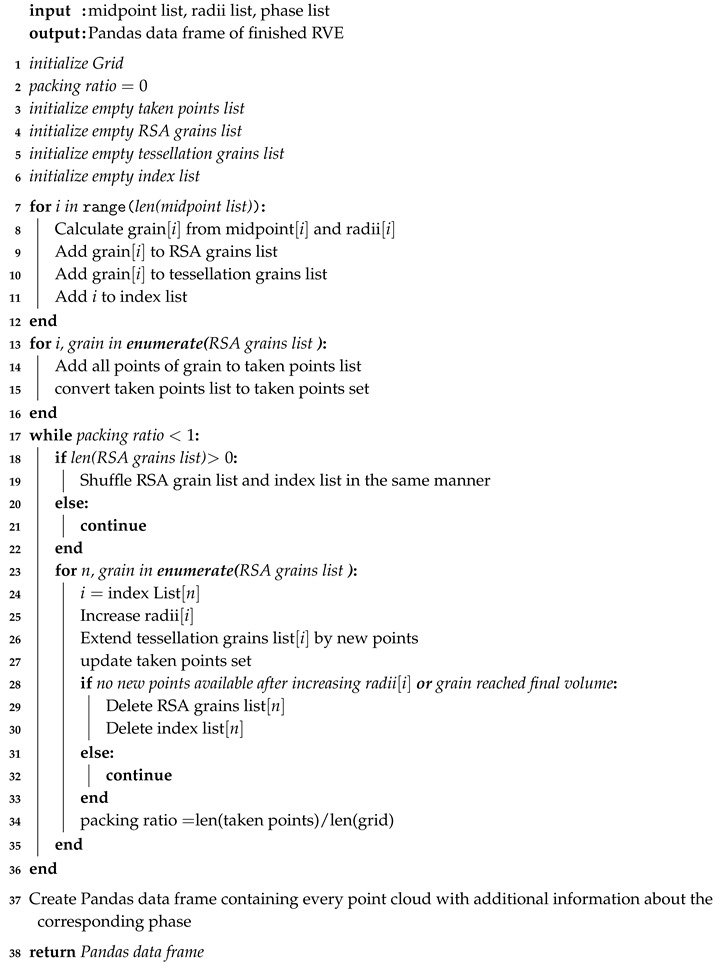


## 3. Results

In this study a set of input parameters for a microstructure was chosen in order to create a synthetic microstructure. The input parameters contain the idealized radii for each grain. Since the given data set is based on EBSD pictures and they are only in 2D the third dimension has to be interpolated. This was done by the assumption that grains can be approximated by ellipsoids build up with the radii *a*, *b* and *c*. The shape of an ellipsoid with its radii is schematically shown in Figure 4.

Since there is no existing characterization for the materials microstructure in 3D, an assumption had to be derived from metallographic pictures as a first initial guess. The metallographic pictures are shown in Figure 5. So to summarize, the data for the grain size and two dimensional shape was taken from EBSD data while the assumption for the three dimensional shape that a=b was taken from the LOM analysis.

The three cross sections in Figure 5 show that a=b is a promising assumption for the ellipsoids. Due to this assumption an input parameter set for these kind of ellipsoids was created. The log probability density of the grain size described by the ellipsoid volume for each grain is shown in Figure 6.

This parameter set, created by the assumptions mentioned above contains 2049 grains that fill a volume of (61×61×61)μm${}^{3}=226,981\;\mu$m${}^{3}$. The output result of the new RVE-generator is shown in Figure 7.

For analyzing the final RVE shown in Figure 7 the grain size of each grain in the RVE was extracted. An evaluation via KDE led to the log probability density shown in Figure 8 as result. The figure also shows a peak for grains with a volume around 10 μm${}^{3}$. The evaluated RVE for Figure 8 was generated with a grid with an edge length of 61 μm. The grid contains 96×96×96=884,736 points. This number can be changed as desired and is only limited by calculation resources of the computer. The packing ratio for the RSA algorithm was 20% which means that the ellipsoids placed in the volume during the RSA algorithm occupied 20% of the given space. The other 80% were then taken by the controlled growth function. It must be mentioned here that the packing ratio of the RSA algorithm does not play such an important role as it does if a voronoi tessellation is used. The here used discrete tessellation function controls the volume of each grain. Therefore, grains do not tend to grow too big.

Additionally an RVE with one band of a second phase was created for demonstration purposes. The validation of the band is not yet possible, since there is no clear characterization how a martensite band should be defined. In this demonstration case the band was simply defined as a fixed volume which must contain at least 60% of martensite. The resulting RVE is shown in Figure 9.

## 4. Discussion

The results of this study show, that the RVE generator invented at Integrity of Materials and Structures (IMS) is able of reconstructing microstructures from a given input data set. Although, the results shown so far only consider the grain size at this stage the agreement of input data and output data is very promising. Features like slope, grain orientation as well as misorientation will be implemented in the future. Additionally to separate grains and their morphology this generator is also able to reconstruct banded microstructures. In order to make this generator more powerful it is planned to implement a module which will create an FE mesh from the grid. This could be done with the scikit-image python library. A very promising package to realize such an aproach seems to be the marching cubes lewiner package within this library. This package uses the method from Lewiner et al. to calculate the iso-surface from a given volumetric data set, for example, point clouds converted into voxel data [31]. The iso-surface and the volume enclosed by the surface can then be meshed with mesh generators like GMSH (GMSH version 4.5.6) [32].

With this module it would then be possible to run FE simulations on these RVEs and damage could be observed with the corresponding material models.

Considering the over all range of all grain sizes the input and output distribution have a very good agreement. When taking a closer look at the range where most of the grain volumes are located as shown in Figure 10b a slight offset between the two peaks appears. This result tends to be underestimated because it clearly shows an offset of the peaks. However, it must be considered that this is a difference of around 2μm${}^{3}$ in a total volume of 226,981μm${}^{3}$ and in addition to that the figure shows a very robust log probability density not a regular log normal distribution which could easily be fitted perfect on the input data. The offset is caused by the discretization of the volume. For ellipsoids a discrete volume is in general smaller than the continuous volume of the same ellipsoid the reason for this is exemplary shown for a two dimensional case in Figure 11.

While the volume of the ellipse is analytically $V_{ell}=96.2$ cm${}^{2}$ the discrete volume only reaches 92 cm${}^{2}$. This also counts for three dimensions. However, this problem can be decreased by increasing the number of points within the grid. It was observed that grids with higher numbers of points led to better results. This can be seen in Figure 12 where the peaks of the two curves move closer to each other.

## 5. Conclusions

In this study it was discussed that common RVE-generators are lacking some features that are essential when ductile damage or fatigue crack initiation is to be described on the basis of microstructural simulations. Such as elongated grains, the slope of grains or banding in secondary phases. The newly introduced RVE-generator is capable of successfully reconstructing the grain morphology, elongated grains and banded microstructures. Therefore, the generated RVEs can be used for a microstructure optimization based on local characteristics of the real microstructure. All the implemented features are based on Python-modules and can be easily extended or replaced when other features are desired. A module for the consideration of nonmetallic inclusion could be implemented in a similar manner as the banding of the microstructure. Also, it would be entirely conceivable, to expand the generator into the field of fiber composites, ceramic composites or polycrystalline rocks and minerals. Also, it is planned for the future to implement grain orientations and the slope of the grains. Additionally to that a module for FE-meshing will be implemented.

## Figures and Tables

**Figure 1 materials-13-01887-f001:**
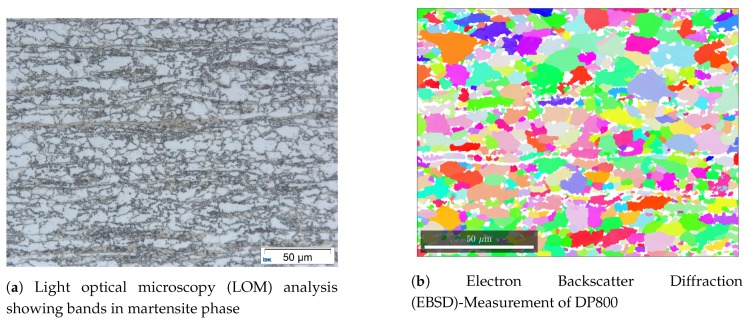
Microstructure analysis of DP800.

**Figure 2 materials-13-01887-f002:**
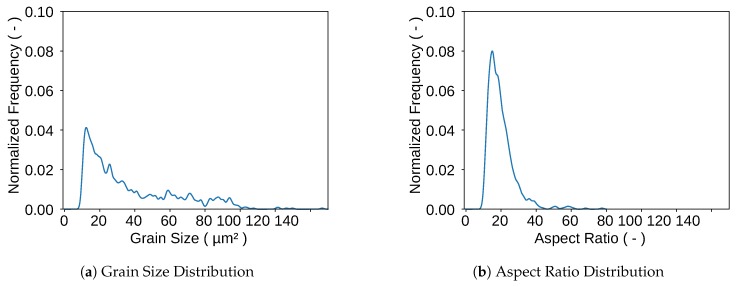
Distributions of Grain Size and Aspect Ratio used as Input data for the Representative Volume Element (RVE)-Generation.

**Figure 3 materials-13-01887-f003:**
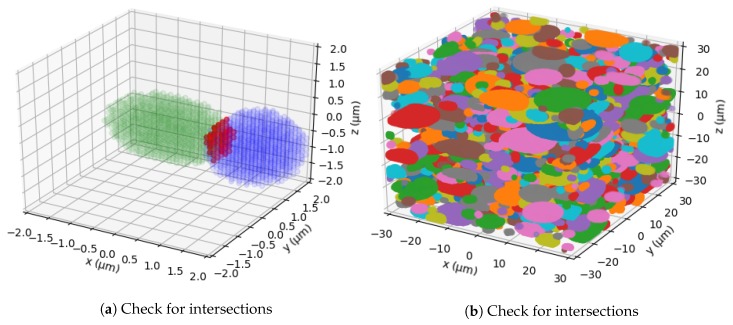
Stages of Random Sequential Addition (RSA).

**Figure 4 materials-13-01887-f004:**
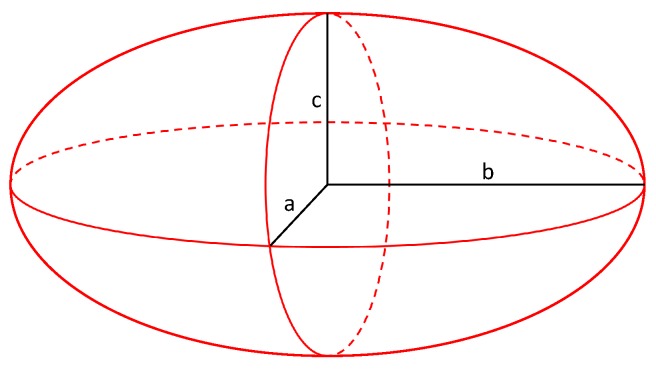
Schematic figure of an ellipsoid with the different radii *a*, *b* and *c*. For modeling the microstructures it was assumed that a=b.

**Figure 5 materials-13-01887-f005:**
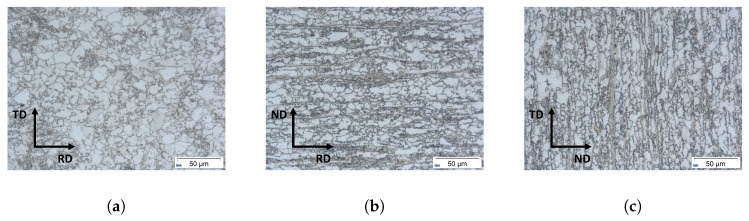
Three different crosssections of the materials microstructures show that it can be assumed that ellipsoids with *a* = *b* and *a* ≠ *c* describes the microstructure better than regular spheres. The three crosssections show metallographic pictures of the Microstructure in the TD x RD-Plane (**a**), the ND × RD-Plane (**b**) and the ND × TD-Plane in (**c**).

**Figure 6 materials-13-01887-f006:**
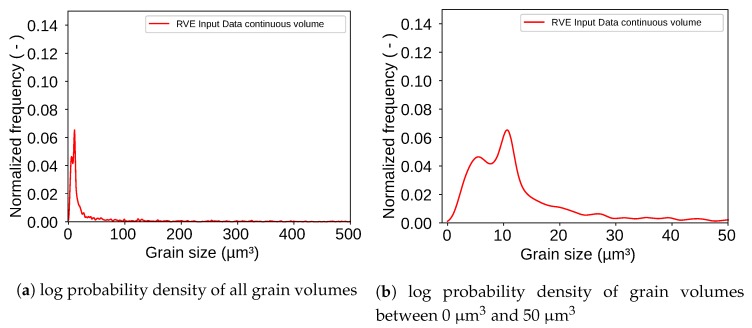
Log probability density of grain volumes shows that most grain volumes are in the range between 0 μm^3^ and 50 μm^3^.

**Figure 7 materials-13-01887-f007:**
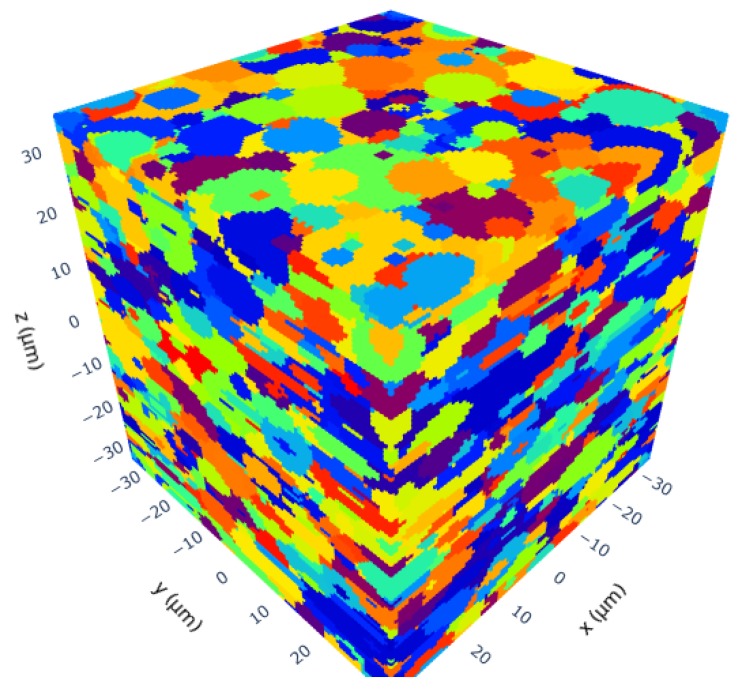
Final result after the discrete tessellation.

**Figure 8 materials-13-01887-f008:**
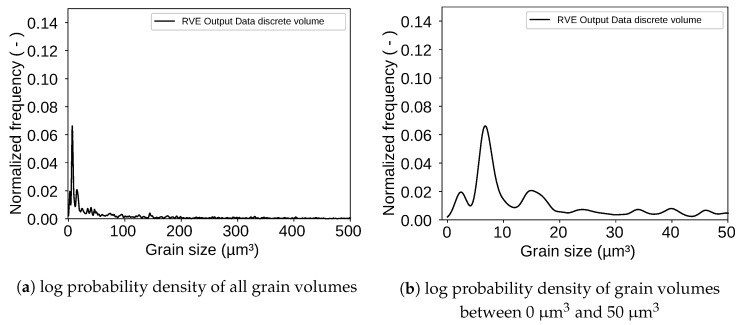
Log probability density of grain volumes shows that most grain volumes are in the range between 0 μm^3^ and 50 μm^3^.

**Figure 9 materials-13-01887-f009:**
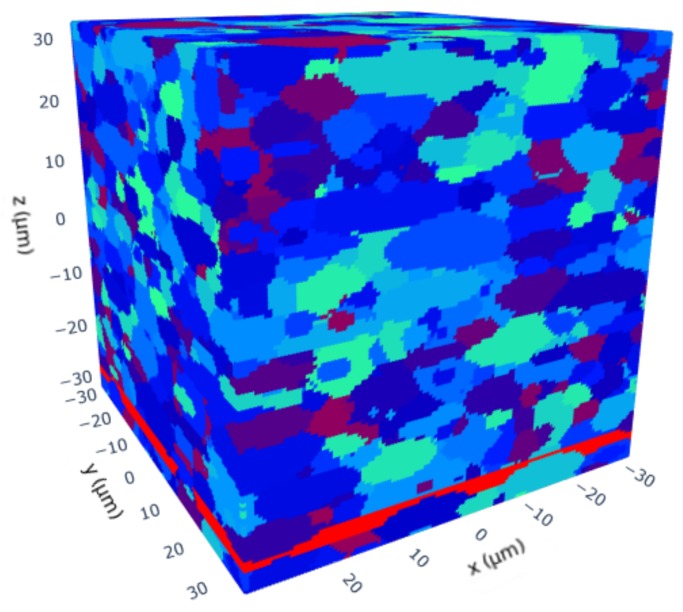
RVE with a martensite band shown in red color. The martensite band is defined as a volume within the cube that must contain at least 60% of martensite.

**Figure 10 materials-13-01887-f010:**
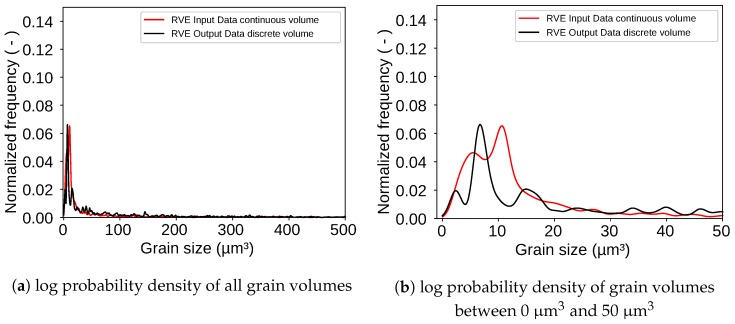
Log probability density of grain volumes shows that most grain volumes are in the range between 0 μm^3^ and 50 μm^3^.

**Figure 11 materials-13-01887-f011:**
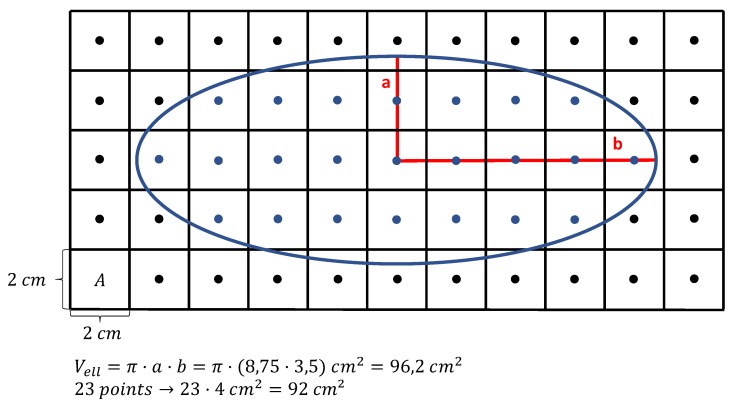
Schematic drawing of a 2D ellipse in a discrete and continuous volume. It is shown that the continuous volume for ellipses and therefore also for ellipsoids is in general bigger than the discrete volume.

**Figure 12 materials-13-01887-f012:**
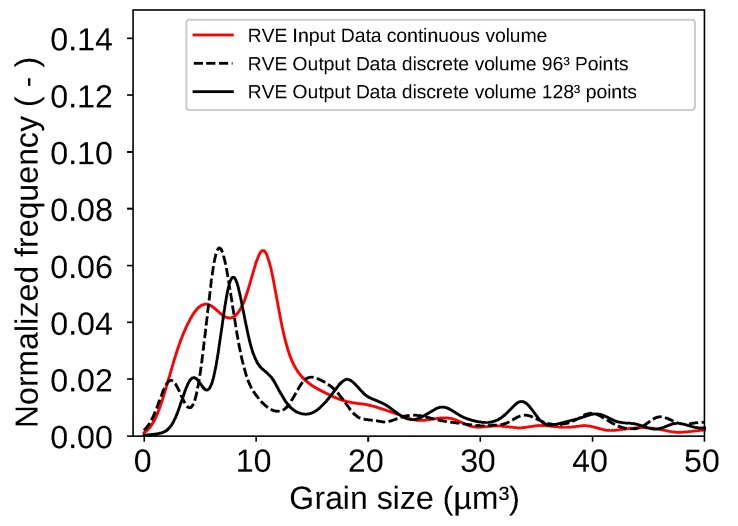
Log probability density of the input values as well as the output values for an RVE created with 128×128×128 points.

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
