# Peer review of "A Novel Approach to Discrete Representative Volume Element Automation and Generation-DRAGen"

_materials, 2020, doi:10.3390/ma13081887_

Round 1
Reviewer 1 Report
The work is very interesting for metallurgy field and is worth considering this paper for this journal. However, some clarification are need.
Font size of Figure 2,3, 7 is too small
From Figure 2 is assumed that the majority of grain size is of 2.5 um or even less; however, Figure 1 b reveals the sizes much higher (at list 10um). So, it is quite difficult to assume “Kernel Density Estimation (KDE) for the grain size is much more trusted”. Besides, may a function like ellipsoids is not very necessary fit for solving the RVE as the grain geometry is quite complex.
Also what about nonindexed data that are close to grain boundary in your case
What about if there is continuum recrystallization so, do now work with this assumption “So, whenever a grain reaches its predefined maximum volume the growth function”
OK, to think about crack initiation “local events like fatigue crack initiation caused by local properties of the microstructure.” But there needs a threshold load …
“Therefore, grains don’t tend to grow too big” to prove this are two possibility to check against literature or make some experiments
It is little bit misleading that you said you capture the data from EBSD while you describe this examples by OM as per Figure 5
Author Response
Dear reviewer,
thank you very much for the useful and very fast review of our manuscript. Below we listed all your comments and questions with specific answers on each of them. We hope that with these answers and the and subsequent modifications of the manuscript everything is to your complete satisfaction.
We wish you all the best during this difficult time - stay healthy!
With kind regards,
the authors
[1] Font size of Figure 2,3, 7 is too small
Response:
The authors would like to thank the reviewer for this useful comment. Therefore, all three figures have been enhanced for better readability.
[2] From Figure 2 is assumed that the majority of grain size is of 2.5 um or even less; however, Figure 1 b reveals the sizes much higher (at list 10um). So, it is quite difficult to assume “Kernel Density Estimation (KDE) for the grain size is much more trusted”. Besides, may a function like ellipsoids is not very necessary fit for solving the RVE as the grain geometry is quite complex.
Response:
The authors would like to thank the reviewer for this important point. We realized that the dataset behind both graphs in Fig 2 did not belong to the EBSD Measurement in Fig. 1 b. Therefore, we plotted them again with the correct dataset and replaced the figure with the correct plot.
[3] Also, what about nonindexed data that are close to grain boundary in your case
Response:
The authors would like to thank the reviewer for this note, we feel that the passage was not written informative enough. Therefore, we added the following sentence to the passage in section 3: “The nonindexed area, depicted by the white color, is assumed to be 100 % martensite.” It can be found in line 160-161
Martensite however in this study is only considered, when adding martensite bands to the RVE as discussed in section 3 and shown in Fig. 9.
[4] What about if there is continuum recrystallization so, do now work with this assumption “So, whenever a grain reaches its predefined maximum volume the growth function
Response:
The authors would like to thank the reviewer for this remark. The reviewer’s comment is correct, our algorithms are not capable of simulating the formation process of the microstructure or any kind of recrystallization. Therefore, the microstructure was analyzed after the whole production process. This microstructure that exists in the material after the production is then to be reproduced by our RVE-Generator. The growth function in this case is simply a method to fill up the spaces between the ellipsoids from the RSA-Algorithm. This function however, needs a control mechanism in order to reproduce the correct grainsize distribution.
Yet, we feel that our formulation is a little bit misleading. When we talk about grains in the RVE we mean the point clouds representing grains in the final microstructure of the material. Grain-growth is therefore to be understood as a growth of point clouds into the predefined volume of the RVE. To emphasize this and avoid misunderstandings the following sentences were added in line 197-202: “It must be noted here that the goal of the generator is to reconstruct a target microstructure. In the following point clouds will also be referred to as grains and the growth of point clouds will be called grain growth. These expressions must not be understood in any sense of metallurgic grain growth or similar. The following description refers only to point clouds developing into their target shape and target size in order to depict the grains of the target microstructure correctly”
[5] OK, to think about crack initiation “local events like fatigue crack initiation caused by local properties of the microstructure.” But there needs a threshold load
Response:
The authors would like to thank the reviewer for this useful comment. We changed the above-mentioned sentence to the following: “In order to investigate local events like fatigue crack initiation caused by local properties of the microstructure, martensite bands must be modeled separately to be able to define needed failure criteria only in the region of the martensite bands”. This can be found in line 273-275.
[6] “Therefore, grains don’t tend to grow too big” to prove this are two possibility to check against literature or make some experiments
Response:
The authors would like to thank the reviewer for this remark. Again, we feel that this comment was made due to some misleading formulations. And the changes in line 197-202 that were mentioned above explains in which way the grain growth is to be understood.
[7] It is little bit misleading that you said you capture the data from EBSD while you describe this examples by OM as per Figure 5
Response:
The authors would like to thank the reviewer for this useful comment. We added the following sentence to avoid any misunderstanding at this point.: “So to summarize, the data for the grain size and two-dimensional shape was taken from EBSD data while the assumption for the three-dimensional shape that a = b was taken from the LOM analysis.”. This can be found in line 295-297
Reviewer 2 Report
This paper develops and implements methods for generating numerical RVEs, here used primarily to represent microstructure statistics of martensitic steels. The methods could be applied to other polycrystals, in general. The methods advocated here account for important and complex microstructure features such as non-spherical entities, banding of second phases, and inclusions. The algorithms are discussed in detail, along with application to a DP800 steel.
The methods and results presented in this work advance the state-of-the-art and contain features apparently not available in other softwares such as DREAM.3D, for example. The writing and figures are of high quality. The paper should be of interest to materials scientists working on computational rendering and computational solid mechanics of microstructures. A few minor revisions might be considered:
[1] It can be noted that the RVE-generation methods could be used to model ceramics and ceramic composites, polycrystalline rocks and minerals, in addition to metals.
[2] Some references and brief discussion on how FE meshes can be assigned to the microstructures should be added.
[3] Also, for completeness, some references on computational solid mechanics models of polycrystals, including for example continuum elasticity, plasticity, and phase-field models (elasticity, plastic slip, twinning, fracture, heat conduction, etc.) should be included in the introduction. These would provide context for how the microstructure meshes would eventually be used by engineers.
Author Response
Dear reviewer,
thank you very much for the useful and very fast review of our manuscript. Below we listed all your comments and questions with specific answers on each of them. We hope that with these answers and the and subsequent modifications of the manuscript everything is to your complete satisfaction.
We wish you all the best during this difficult time - stay healthy!
With kind regards,
the authors
[1] It can be noted that the RVE-generation methods could be used to model ceramics and ceramic composites, polycrystalline rocks and minerals, in addition to metals.
Response: The authors would like to thank the reviewer for this interesting remark. We added this point in section 5 as an outlook for future developments: “Also, it would be entirely conceivable, to expand the generator into, e.g. the field of fiber composites, ceramic composites or polycrystalline rocks and minerals.” This can be found in line 356-358
[2] Some references and brief discussion on how FE meshes can be assigned to the microstructures should be added.
Response: The authors would like to thank the reviewer for this useful comment. Some ideas how to realize the point to mesh conversion were added in section 4: “This could be done with the scikit-image python library. A very promising package to realize such an approach seems to be the marching cubes lewiner package within this library. This package uses the method from Lewiner et al. to calculate the iso-surface from a given volumetric data set, e.g. point clouds converted into voxel data [26]. The iso-surface and the volume enclosed by the surface can then be meshed with mesh generators like GMSH [27]”. This can be found in line 326-330
[3] Also, for completeness, some references on computational solid mechanics models of polycrystals, including for example continuum elasticity, plasticity, and phase-field models (elasticity, plastic slip, twinning, fracture, heat conduction, etc.) should be included in the introduction. These would provide context for how the microstructure meshes would eventually be used by engineers.
Response: The authors would like to thank the reviewer for this useful hint. We added the following passage to the introduction:
“Material models that can be used with the FE² method are e.g. the Modified Bai–Wierzbicki plasticity model (MBW) as described by Lian et al. and Wu et al. [12, 13] . Another type of Material models are Crystal Plasticity models (CP) as they were used by e.g. Cruzado et al. or Segurado et al. [14, 15]. A quite detailed summary of other constitutive laws, kinematics, homogenization and multiscale methods is given by Rothers et al. [16].” This can be found in line 38-43